# Therapeutic Use of Intranasal Glucagon: Resolution of Hypoglycemia

**DOI:** 10.3390/ijms20153646

**Published:** 2019-07-25

**Authors:** Antonio E. Pontiroli, Elena Tagliabue

**Affiliations:** 1Dipartimento di Scienze della Salute, Università degli Studi di Milano, Ospedale San Paolo, Via Antonio di Rudinì 8, 20142 Milan, Italy; 2IRCCS Multimedica, Servizio Statistico, 20138 Milan, Italy

**Keywords:** hypoglycemia, diabetes mellitus, insulin, glucagon, intranasal

## Abstract

Episodes of hypoglycemia are frequent in patients with diabetes treated with insulin or sulphonylureas. Hypoglycemia can lead to severe acute complications, and, as such, both prevention and treatment of hypoglycemia are important for the well-being of patients with diabetes. The experience of hypoglycemia also leads to fear of hypoglycemia, that in turn can limit optimal glycemic control in patients, especially with type 1 diabetes. Treatment of hypoglycemia is still based on administration of carbohydrates (oral or parenteral according to the level of consciousness) or of glucagon (intramuscular or subcutaneous injection). In 1983, it was shown for the first time that intranasal (IN) glucagon drops (with sodium glycocholate as a promoter) increase blood glucose levels in healthy volunteers. During the following decade, several authors showed the efficacy of IN glucagon (drops, powders, and sprays) to resolve hypoglycemia in normal volunteers and in patients with diabetes, both adults and children. Only in 2010, based on evaluation of patients’ beliefs and patients’ expectations, a canadian pharmaceutical company (Locemia Solutions, Montreal, Canada) reinitiated efforts to develop glucagon for IN administration. The project has been continued by Eli Lilly, that is seeking to obtain registration in order to make IN glucagon available to insulin users (children and adolescents) worldwide. IN glucagon is as effective as injectable glucagon, and devoid of most of the technical difficulties associated with administration of injectable glucagon. IN glucagon appears to represent a major breakthrough in the treatment of severe hypoglycemia in insulin-treated patients with diabetes, both children and adults.

## 1. Hypoglycemia in Patients with Type 1 and Type 2 Diabetes, a Risk for Major Complications

Treatment of diabetes mellitus has substantially changed over the last 20 years for both patients with type 1 (T1D) and type 2 (T2D) diabetes. Older oral antidiabetic agents (metformin, sulphonylureas, glinides, and glitazones) now are accompanied by oral (e.g., dipeptidyl–peptidase (DPP-IV) antagonists, sodium-glucose cotransporter (SGLT-2) inhibitors) and injectable (e.g., glucagon-like peptide (GLP-1) agonists drugs), and insulin analogs (with an extremely short or an extremely long duration of action) are now in use. Control of hyperglycemia is now possible for virtually all patients with diabetes, but an unmet need is still represented by prevention and treatment of hypoglycemia; euglycemia without acute metabolic complications, still remains a major goal of management in patients with diabetes [1].

Hypoglycemia, symptomatic hypoglycemia, and severe hypoglycemia (SH, a circumstance where the patient is unconscious and requires the assistance of someone else) are frequent in T1D and T2D patients who use insulin or in T2D patients on sulphonylureas. Among insulin-treated patients the frequency of hypoglycemia is greater in T1D than in T2D patients, and depends on intensive insulin treatment, on regimens of insulin administration, and on age [2,3,4,5,6,7,8,9,10,11].

The experience of hypoglycemia leads to a fear of hypoglycemia, that in turn can limit optimal glycemic control in many children and adolescents, as well as in adults with type 1 diabetes [12,13]. More importantly, hypoglycemia can lead to cardiovascular accidents [14], falls and trauma [15], and cognitive impairment [16]. Both prevention and treatment of hypoglycemia are important for the well-being of patients with diabetes [1]. Prevention is mostly performed through education on optimal timing of food or snack ingestion and exercise [17], blood glucose monitoring, and management of hypoglycemia unawareness [18]; in fact, frequent and recurrent hypoglycemia in patients with type 1 diabetes can lead to hypoglycemia unawareness, which puts individuals (more adult than pediatric patients) at greater risk of continued and severe hypoglycemia. Treatment of hypoglycemia is still based on administration of carbohydrates (oral or parenteral according to the level of consciousness) or of glucagon (intramuscular (IM) or subcutaneous (SC) injection) [19]. Glucagon, in particular, is a peptide hormone, and as such, carries the challenges associated with stability and with all the limitations common to administration of most peptides.

## 2. Intranasal Administration of Peptide Hormones

Peptide hormones are traditionally administered by the parenteral route, be it by SC, IM or intravenous (IV) injection, as oral administration is precluded by digestion and inactivation in the gastrointestinal tract [20]. Attempts to identify alternative routes of administration have been tested since the early 1920s, in particular for hormones that require life-long treatments, such as insulin [21,22]. Peptide hormones greater than 10 amino acids require promoters to be absorbed intranasally (IN); several strategies have been tested to improve IN absorption of peptides, like the use of bile salts, beta-cyclodextrins, or detergents, or the use of more refined technologies, like the use of nanoparticles, liposomes, and bioadhesive microspheres [23]. IN administration is now established for a few hormones such as desmopressin, oxytocin, luteinizing hormone-releasing hormone (LHRH) and its analogues (buserelin, leuprolide, nafarelin) [24], salmon calcitonin [25]. In contrast, IN administration of insulin, growth hormone-releasing hormone (GHRH), GH, corticotropin-releasing hormone (CRH), human calcitonin, somatostatin, and hexarelin has been without success so far [21,23]. IN administration has also been used to deliver PHs directly to the brain. Among other effects, in laboratory animals, intranasally administered insulin has been shown to exert effects opposite to parenteral insulin administration. This means that insulin in the brain regulates peripheral insulin sensitivity [26], and accordingly, IN insulin induces insulin sensitivity in lean and obese subjects [27]. Finally, intranasal insulin administration has been used to induce immune tolerance in T1D subjects [28], and GLP-1 has been administered IN to modify learning and for neuroprotection [29,30].

## 3. Intranasal Administration of Glucagon

It was shown for the first time in 1983 that IN glucagon drops (with sodium glycocholate as a promoter) was able to increase blood glucose levels in healthy volunteers [31]. A few years later it was shown that IN glucagon solutions and IN glucagon powders are similarly effective, provided a promoter is used, as IN glucagon without promoters is not absorbed through the nose and is without effects whatsoever [32,33,34,35]. Several promoters have been used by different investigators (i.e., sodium-glycocholate, 9-lauryl-ether, deoxycholic-acid, didecanoyl-phosphatidylcholine (DDPC) with alpha-cyclodextrine, sodium caprate, and microcrystalline cellulose [32,33,34,35]). During that decade, several authors showed the efficacy of IN glucagon to resolve hypoglycemia in normal volunteers and in patients with diabetes, both adults and children [36,37].

In spite of these results, at that time, the pharmaceutical industry did not take the opportunity to develop intranasal glucagon. We can assume that the industry of that period considered that IM glucagon was sufficiently effective [38], and that the market opportunity might not justify the investment required to investigate alternative routes of administration that would be better suited to the intended user, and therefore why try alternative routes of administration with the risk of ineffectiveness and the risk of legal compensation issues.

Only in 2010, based on the evaluation of patients’ beliefs and expectations [39,40], a canadian pharmaceutical company (Locemia Solutions, Montreal, Canada) retrieved old data and reinitiated efforts to develop glucagon for IN administration. The company recognized that the ultimate user of glucagon was likely much more comfortable administering a needle-free formulation and, on that basis, conducted a series of safety, efficacy, and human factor studies to support registration of the product [39,40]. The project has been continued by Eli Lilly, a U.S. pharmaceutical industry that is seeking to obtain registration in order to make IN glucagon available to insulin users (children and adolescents) worldwide.

## 4. Biology of Intranasal Glucagon

In contrast to other peptide hormones (e.g., insulin), glucagon does not show a clear dose-response relationship, suggesting that the glycemic response to glucagon is saturable. In fact, after IV or IM glucagon administration, there is a clear relationship between doses administered and pharmacokinetic parameters (C_max_, etc), but progressively increasing doses do not result in dose-related responses of glucose (NDA 020928, Eli Lilly).

IN glucagon drops, powders, and sprays increase blood glucose levels in healthy volunteers and in patients with diabetes, both under normal circumstances and during hypoglycemia [31,32,33,34,35,36,37,38]. Aside from effects on glucose metabolism, IN glucagon exerts other, as yet poorly investigated, actions. For example, IN glucagon induces gastric hypotonia [41] and (0.7 mg) acutely increases energy expenditure without inducing hyperglycemia in overweight/obese adults [42] with no effect on food intake and appetite.

The new formulation now under investigation differs from the pioneer formulations. It is a dry powder formulation (with beta-cyclodextrin plus dodecylphosphocholine as the promoter) and is administered using a simple, handheld, nasal applicator [43,44].

The bioavailability of peptides administered is lower after IN than after parenteral administration. The same applies to IN glucagon, and glucagon bioavailability is lower after IN glucagon than IM glucagon, resulting in lower peak plasma glucagon concentrations, however, IN dosing results in a glycemic excursion that is similar to IM glucagon in terms of return to normal glucose levels. IN glucagon (3 mg) has been shown to be noninferior to IM glucagon (1 mg) [45] in treating insulin-induced hypoglycemia in adults with T1D. A recent study compared 2 mg and 3 mg doses of IN glucagon to weight-based IM glucagon (0.5 mg or 1 mg) in pediatric patients (4–17 years old) with T1D [46]. This study found that both the 2 mg and 3 mg IN dose levels were well tolerated and resulted in a similar glycemic response pattern to weight-based IM glucagon. As such, it appears that a single dose level (i.e., 3 mg) can be used in children and adults with T1D, which would simplify prescribing this medication across the entire age range of patients with T1D (i.e., 4 years of age or older). In other studies, IN glucagon was similarly well tolerated after a single 3 mg dose, or after 2 doses administered with a 15 min interval (i.e., 6 mg total) or 2 doses administered together [47]. The bulk of these data indicates that IN glucagon administration is feasible, and that the glucose response to IN glucagon is saturable, as happens with IM glucagon. A meta-analysis showed that IN and IM glucagon are similarly effective, with no differences in terms of success [38].

## 5. Possible Advantages of Intranasal Glucagon

### 5.1. Technical Skills for Preparing Injectable Glucagon

Glucagon solutions are not stable, owing to the propensity of glucagon to form fibrils once in an aqueous solution [48]. As a result, as long as stable glucagon solutions are not available [49], currently available glucagon emergency kits require reconstitution of lyophilized powder in a diluent immediately prior to IM injection by family members or others who may not be well trained in or comfortable with giving injections [50]. Injection of IM glucagon is thought to be so complex that many U.S. states allow only nurses or other trained health professionals to give glucagon injections while youngsters with T1D are in school [51].

### 5.2. What Are Patients Expectations

In a survey conducted by Glu, on the T1D exchange patient/caregiver online community (myglu.org), nearly 75% of participants stated they never or rarely carried a glucagon emergency kit [50]. Several studies have demonstrated the difficulty associated with the administration of currently available injectable glucagon preparations. A survey, which was conducted in 1997 [39], ascertained opinions on the currently available glucagon emergency kits among patients with type 1 diabetes; the majority (66%) of the patients stated they would prefer an IN administered glucagon, if and when available, and 82% believed that family members, teachers, and colleagues would prefer to administer emergency therapy by the IN route for treating severe hypoglycemia. From this viewpoint, results reported by Yale and coworkers on faster use and fewer failures with needle-free nasal glucagon versus injectable glucagon in severe hypoglycemic rescue, are of great interest [52].

### 5.3. A Question about Common Cold and Nasal Decongestants

One of the most common questions asked about IN administration of drugs, in general, and about IN glucagon, in particular, is, what happens in a patient with a common cold or nasal congestion. This concern was addressed by Guzman and coworkers, who found that nasal congestion associated with the common cold, with or without nasal decongestants, does not impair glucose response to IN glucagon [53].

### 5.4. Resolution of Hypoglycemia Begins Almost Immediately with Both IM and IN Glucagon

The glucose response to IN glucagon is very rapid, as it is with IM glucagon; direct comparisons in the same subjects yielded superimposable kinetic and dynamic curves for glucagon and glucose, respectively. The only difference was, in some studies, a 1 min lag in glucose response to IN glucagon, as compared to IM glucagon, in some [45] but not in all studies [46]. Given the other advantages of IN glucagon, this difference seems negligible and, in any case, reassuring.

### 5.5. What Happens if an Administration Fails? Failure of IM Glucagon and of IN Glucagon

During the 1970s and the 1990s, there were concerns about failures of treatment of hypoglycemia with the remedies available at that time, i.e., IV dextrose and IM glucagon. Aside from one study [54], failures with either IM glucagon or dextrose were very few, and the two remedies were superimposable as to effectiveness [38]. The data available at this time indicate that once administered IN and IM glucagon are similarly effective, and that failures with IN glucagon have been fewer than with IM glucagon [52], and a meta-analysis showed a similar effectiveness of IN and of IM glucagon [38]. Importantly, however, studies clearly demonstrate that IN glucagon administration is much more easily accomplished by the user, resulting in much higher treatment success with IN than with injected glucagon [52]. As such, it appears that IN glucagon is as effective as an injected glucagon, and more easily administered. The real failures are probably extremely rare, and it seems wise to suggest that, in the case of real failure of IN glucagon or of injectable glucagon, a second dose is likely to be effective.

### 5.6. The Real World and the Role of Caregivers

There are reasons to suggest that, in real-world settings, IN glucagon should be effective, safe, and easy to use, as suggested by controlled studies. The ease of use and the efficacy of IN glucagon reported in controlled or simulated studies have been confirmed in clinical studies where IN glucagon was administered to T1D adults and caregivers of T1D children to treat acute hypoglycemia episodes at home and in school settings [55,56]. Needle-free, ready-to-use IN glucagon could represent a particularly strong alternative to injectable glucagon for caregivers, family members, friends, and colleagues who may someday face the difficult task of treating SH in children using insulin or adults with T1D. It is important to note that glucagon is truly unique in that it is prescribed for insulin users but it is actually intended to be administered in a moment of severe stress (i.e., an episode of severe hypoglycemia) by a third party (i.e., a caregiver) who may not have any familiarity with injection technique.

The economic impact of the usability advantage of IN glucagon over IM glucagon was explored in cost offset and budget impact analyses for the US setting, for T1D and T2D patients treated with basal–bolus insulin regimens. Reduced spending resulted from reduced professional emergency services utilization as successful treatment was more likely with IN glucagon [57], and therefore IN glucagon has the potential to improve hypoglycemia emergency care and reduce severe hypoglycemia-related treatment costs. The above considerations are valid, of course, for the currently recognized evidence for IN glucagon, i.e., severe hypoglycemia (SH). One might imagine a different scenario, such as, due to the ease of administration, and also of self-administration, some people may administer IN glucagon to themselves to treat an episode of hypoglycemia before it becomes too serious (SH).

## 6. Limitations

The frequency of hypoglycemia and of severe hypoglycemia is quite frequent and represents a real burden for T1D patients. In 2011, hypoglycemia accounted for approximately 282,000 emergency department visits by adults with diabetes [58]. Glucagon is as effective as glucose in the resolution of hypoglycemia [38,59], but its use is infrequent [60,61]. Of interest is the evidence that, among insulin-treated patients, the frequency of access to emergency department because of hypoglycemia was lower for patients with prescribed glucagon than that of other patients [61]. Possible reasons for the low diffusion of glucagon prescriptions have been discussed before in [39,50], however, it remains to be seen whether ease of use expands the use of glucagon for hypoglycemia. Difference in price between injectable and IN glucagon is unclear at this time. The potential for higher sales of IN glucagon because of its ease of use may potentially provide cost savings. There is also a potential for companies who sell injectable glucagon to provide more cost savings once the IN glucagon becomes available for those starting or already taking injectable glucagon.

## 7. Conclusions

Glucagon, both IM and IN, is intended for use by another person since, by definition, treatment of SH requires the assistance of another person. Outcomes depend not only on the efficacy of the product, but, more importantly, on the ability of a caregiver, or another person, to actually administer the drug. Intranasal glucagon appears to represent a major breakthrough in the treatment of severe hypoglycemia in insulin-treated patients with diabetes, both children and adults [58,59], as it represents a more effective route of administration for caregivers that face the challenge of dealing with the stress of an episode of SH. IN glucagon is as effective as IM glucagon, and devoid of most of the technical difficulties associated with administration of injectable glucagon. Given the demonstrated safety, efficacy, and ease of use, especially for a nonmedical person who must treat an episode of SH, there is hope that needle-free IN glucagon may soon be available to address an important unmet medical need (Table 1). Due to the ease of administration, and also of self-administration, some people may administer IN glucagon to themselves to treat an episode of hypoglycemia before it becomes too serious.

## Figures and Tables

**Table 1 ijms-20-03646-t001:** Hypoglycemia in insulin treated patients with diabetes. The problems, the remedies, and the possible improvements.

Problem and Remedies	Technical Problems	Alternatives/Improvements
Hypoglycemia		
Frequent, dangerous for prognosis and for well-being		
Remedies are glucose (p.o. or parenteral) and glucagon	Glucagon has to be reconstituted and injected IM	Intranasal glucagon (was shown in 1983 to raise blood glucose levels in healthy subjects)
	Reconstitution is difficult for untrained caregivers	Powders that do not need reconstitution and solutions and can be worn in a device
	Poorly utilized because of technical problems	Easy to administer, might be used by caregivers or even self-administered
		It works in the presence of common cold and nasal congestion
		It works in the real-world setting
	Side effects: several and common	Side effects: fewer, only local

Abbreviations: p.o., oral route; IM, intramuscular.

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
