# Peer review of "Therapeutic Use of Intranasal Glucagon: Resolution of Hypoglycemia"

_ijms, 2019, doi:10.3390/ijms20153646_

Round 1

Reviewer 1 Report

This is an interesting review on a topic/field (with clinical relevance) to which the authors have greatly contributed over several years.

Overall, the manuscript would benefit from thorough language editing. Another concern is, that it often reads like a sales pitch for IN glucagon. Please stick to the facts and observations and keep lengthy interpretation to a minimum. At the end of the manuscript, advantages and disadvantages should be summarized together with concluding remarks.

In order to better understand the scope of the problem, could the authors provide numbers for how often glucagon has to be administered to treat acute hypoglycemia. How many cases are there per year in the US or in Europe, for example…?

Line 25 … “resumed old data … “ not proper English. Revise.

Lines 35/36 … t1d and t2d twice in same sentence. Revise.

A large part of section 1 (in particular lines 43 to 47) is identical to the abstract. This is sloppy. Revise or delete section 1.

Lines 97 … “Canadian Pharmaceutical Industry…” Why are pharmaceutical and industry capitalized? Company is likely a better word for industry as there is only one company alluded to in the parentheses.

Line 105 … “at difference …” awkward phrasing. Revise. Also, misuse of i.e.. Should be e.g., as insulin serves as an example for peptide hormones.

Lines 106 to 110… Please make the statements in that section more clear. It’s confusing to read that there is no dose response relationship only to read that there is a clear relationship in the next sentence. These statements pertain to different parameters, but this needs to be revised and the parameters need to be to clearly defined.

Lines 112-115…. It is not clear why this info is relevant. Can we glean any mechanistic insights from this? Either spell out the relevance or delete.

Line 142 … States of what country??

Line 166 … why is the difference “reassuring”. Delete/revise.

Please introduce a limitations section to create more balance and also speculate why the IN approach has not been embraced by the pharma companies in the past.

Author Response

Thank you so much for your comments.

See light blue remarks

Overall, the manuscript would benefit from thorough language editing.  Another concern is, that it often reads like a sales pitch for IN glucagon.

Thank you in particular for this remark. I guess this referee can understand that, when you are involved for 36 years in a field, you can see a problem in a subjective way.

Please stick to the facts and observations and keep lengthy interpretation to a minimum.

I think this was done

At the end of the manuscript, advantages and disadvantages should be summarized together with concluding remarks.

Done

In order to better understand the scope of the problem, could the authors provide numbers for how often glucagon has to be administered to treat acute hypoglycemia. How many cases are there per year in the US or in Europe, for example…?

This now appears under Limitations

Line 25 … “resumed old data … “ not proper English. Revise.

done

Lines 35/36 … t1d and t2d twice in same sentence. Revise.

done

A large part of section 1 (in particular lines 43 to 47) is identical to the abstract. This is sloppy. Revise or delete section 1.

done

Lines 97 … “Canadian Pharmaceutical Industry…” Why are pharmaceutical and industry capitalized? Company is likely a better word for industry as there is only one company alluded to in the parentheses.

Done as suggested

Line 105 … “at difference …” awkward phrasing. Revise. Also, misuse of i.e.. Should be e.g., as insulin serves as an example for peptide hormones.

Done as suggested

Lines 106 to 110… Please make the statements in that section more clear. It’s confusing to read that there is no dose response relationship only to read that there is a clear relationship in the next sentence. These statements pertain to different parameters, but this needs to be revised and the parameters need to be to clearly defined.

Done as suggested

Lines 112-115…. It is not clear why this info is relevant. Can we glean any mechanistic insights from this? Either spell out the relevance or delete.

Just for completeness

Line 142 … States of what country??

done

Line 166 … why is the difference “reassuring”. Delete/revise.

done

Please introduce a limitations section to create more balance and also speculate why the IN approach has not been embraced by the pharma companies in the past.

Done; the speculation on why pharma companies did not challenge the issue in the past is already in lines 90-95

Reviewer 2 Report

Authors described a meaningful review for the therapeutic use of intranasal glucagon preparation. The review article is well written and it is easy to understand overall. The article deserves to be published in the journal. However, I feel the text is somewhat unreadable because a lot of abbreviations are used in the text. I recommend authors will revise the text by avoiding frequent use of abbreviations. Authors should not use any abbreviation except for IN(intranasal), IM(intramuscular), T1D and T2D in the article.

Author Response

Authors described a meaningful review for the therapeutic use of intranasal glucagon preparation. The review article is well written and it is easy to understand overall. The article deserves to be published in the journal. However, I feel the text is somewhat unreadable because a lot of abbreviations are used in the text. I recommend authors will revise the text by avoiding frequent use of abbreviations. Authors should not use any abbreviation except for IN(intranasal), IM(intramuscular), T1D and T2D in the article.

Thank you so much for your comments. All abbreviations are preceded by full-text terms. Some abbreviations have been eliminated

See light blue remarks